# VIDEOGRAIN: MODULATING SPACE-TIME ATTENTION FOR MULTI-GRAINED VIDEO EDITING

**Xiangpeng Yang** [1]    **Linchao Zhu** [2]    **Hehe Fan** [2]    **Yi Yang** [2*]
[1] ReLER Lab, AAII, University of Technology Sydney    [2] ReLER Lab, CCAI, Zhejiang University
Project Page: https://knightyxp.github.io/VideoGrain_project_page

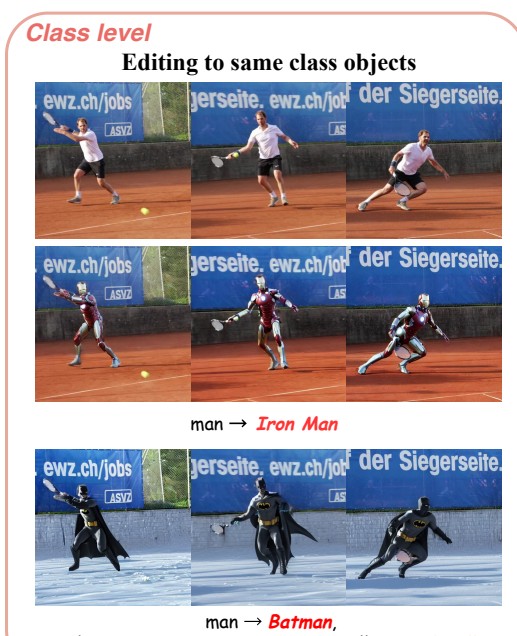

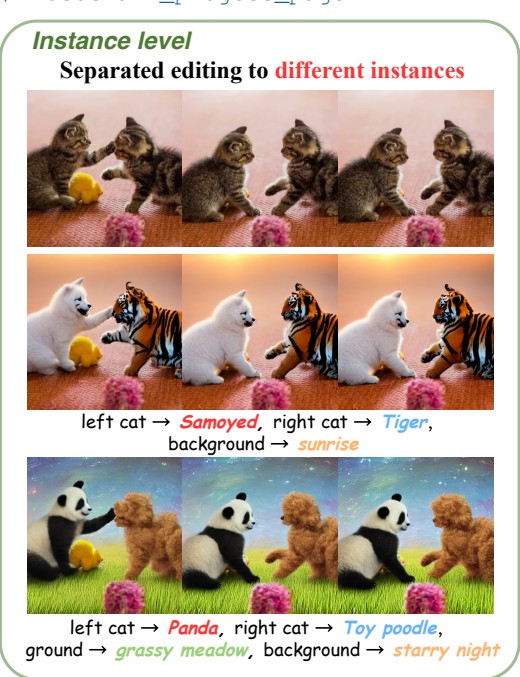

**Class level**
**Editing to same class objects**

man → *Iron Man*

man → *Batman*,
clay court → *snow court*, stone wall → *iced wall*

**Instance level**
**Separated editing to different instances**

left cat → *Samoyed*, right cat → *Tiger*,
background → *sunrise*

left cat → *Panda*, right cat → *Toy poodle*,
ground → *grassy meadow*, background → *starry night*

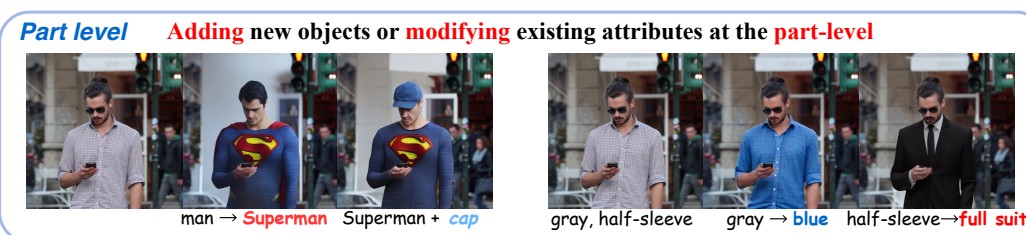

**Part level**    **Adding** new objects or **modifying** existing attributes at the **part-level**

man → **Superman**    Superman + *cap*        gray, half-sleeve    gray → blue    half-sleeve→**full suit**

**Figure 1:** VideoGrain enables multi-grained video editing across class, instance, and part levels.

## ABSTRACT

Recent advancements in diffusion models have significantly improved video generation and editing capabilities. However, multi-grained video editing, which encompasses class-level, instance-level, and part-level modifications, remains a formidable challenge. The major difficulties in multi-grained editing include semantic misalignment of text-to-region control and feature coupling within the diffusion model. To address these difficulties, we present VideoGrain, a zero-shot approach that modulates space-time (cross- and self-) attention mechanisms to achieve fine-grained control over video content. We enhance text-to-region control by amplifying each local prompt's attention to its corresponding spatial-disentangled region while minimizing interactions with irrelevant areas in cross-attention. Additionally, we improve feature separation by increasing intra-region awareness and reducing inter-region interference in self-attention. Extensive experiments demonstrate our method achieves state-of-the-art performance in real-world scenarios. Our code, data, and demos are available on the project page.

*Corresponding author

# 1 INTRODUCTION

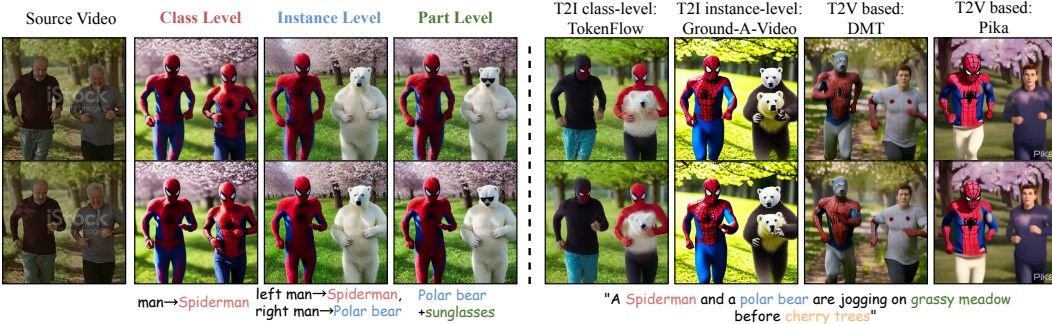

**Figure 2:** Definition of multi-grained video editing and comparison on instance editing

Recent advances in Text-to-Image (T2I) and Text-to-Video (T2V) diffusion models (Rombach et al., 2022; Wang et al., 2023a; Brooks et al., 2024) have enabled video manipulation through natural language prompts. In practical applications, enabling users to edit regions at various levels of granularity based on textual prompts offers greater flexibility. To investigate this, we introduce a new task called multi-grained video editing, which encompasses class-level, instance-level, and part-level editing, as shown in Fig. 2 left. Class-level editing refers to modifying objects within the same class. Instance-level editing means editing different instances into distinct objects. Part-level going further, requires adding new objects or modifying existing attributes at part-level.

While existing methods employ various visual consistency techniques, such as optical flow (Cong et al., 2023; Yang et al., 2023), control signals (Zhang et al., 2023b), or feature correspondence (Geyer et al., 2023). These methods remain instance-agnostic, often mixing features of different instances during editing (see Fig. 2 right). Ground-A-Video (Jeong & Ye, 2023), which inherits text-to-bounding box generation priors (Li et al., 2023), should be instance-level editing but still suffer from artifacts. Similarly, recent T2V-based methods like DMT (Yatim et al., 2024) and Pika (pik), although equipped with video generation priors, struggle with multi-grained edits. We find that the core issue is that diffusion models tend to treat different instances as the same class segments, leading to strong feature coupling across instances, as illustrated in Figure 3.

To address this problem, our primary insight is to 1) enable text-to-region control and 2) keep feature separation between regions. In the typical diffusion models, the cross-attention layer serves as a key component to update textual features control over each spatial region, while the self-attention layer generates globally coherent structures by connecting each frame token across time. Therefore, we propose Spatial-Temporal Layout-Guided Attention (ST-Layout Attn), which modulates both space-time cross- and self-attention in a unified manner to achieve the above goals.

In the *cross-attention layer*, the uniform application of global text prompts across all frame tokens leads to severe semantic misalignment, which reduces the precision of multi-grained text-to-region control. To address this, we modulate cross-attention to amplify each local prompt's focus on its corresponding spatial-disentangled region while suppressing attention to irrelevant areas. In the *self-attention layer*, pixels from one region may attend to outside or similar regions within the same class, leading to feature coupling and texture mixing, which is an inherent limitation of diffusion models that complicates multi-grained video editing. To mitigate this, we modulate self-attention to enhance feature separation by increasing intra-region focus and reducing inter-region interactions, ensuring each query attends only to its target region.

Our key contributions can be summarized as follows:

- To the best of our knowledge, this is the first attempt at multi-grained video editing. Our method enables both class-level, instance-level and part-level editing.

- We propose a novel framework, dubbed *VideoGrain*, which modulates spatial-temporal cross- and self-attention for text-to-region control and feature separation between regions.

- Without tuning any parameters, we achieve state-of-the-art results on existing benchmarks and real-world videos both qualitatively and quantitatively.

## 2 RELATED WORK

### 2.1 TEXT-TO-IMAGE EDITING/GENERATION

In the realm of single attribute text-to-image editing, various approaches have been explored, from manipulating attention maps in Pix2Pix-Zero (Parmar et al., 2023) and Prompt2Prompt (Hertz et al., 2022) to employing masks in DiffEdit (Couairon et al., 2023) and Latent Blend (Avrahami et al., 2022; 2023) for foreground modifications while preserving the background.

For multi-grained editing, efforts like Attention and Excite (Chefer et al., 2023) and DPL (Wang et al., 2023b) focus on maximizing attention scores for each subject token and reducing attention leakage. In image generation, (Kim et al., 2023) modulates attention based on layout masks and dense captions, while (Phung et al., 2023) proposed an attention refocus loss for regularization. However, using single-frame layout masks and dense captioning alone is insufficient for video editing, as it fails to maintain the original video's integrity and temporal consistency.

### 2.2 TEXT-TO-VIDEO EDITING

**Video Editing based on Image Diffusion Models.** Tune-A-Video (TAV) (Wu et al., 2022) is the first work to extend latent diffusion models to the spatial-temporal domain and encode the source motion implicitly by one-shot tuning but still fails to preserve local details. Fatezero (Qi et al., 2023) and Pix2Video (Ceylan et al., 2023) fuse self- or cross-attention maps in the inversion process for temporal consistency. However, (Qi et al., 2023) requires extensive RAM usage and suffers from layout preservation even when equipping TAV for local object editing. (Chai et al., 2023) and (Ouyang et al., 2023), following the Neural Atlas (Kasten et al., 2021) or dynamic Nerf's deformation field (Pumarola et al., 2021), struggle with non-grid human motion. Subsequent methods like Rerender-A-Video (Yang et al., 2023), FLATTEN (Cong et al., 2023) ControlVideo (Zhang et al., 2023b) achieve strict temporal consistency via optical-flow, depth/edge maps, but failed in multi-grained editing while preserving original layouts. Tokenflow (Geyer et al., 2023) enforces a linear mix of nearest key-frame features to ensure consistency but results in detail loss. Ground-A-Video (Jeong & Ye, 2023) leverages groundings for multi-grained editing, but it suffers from feature mixing when bounding boxes overlap.

**Video Editing based on Video Diffusion Models.** Previous video editing work primarily utilized text-to-image SD model (Rombach et al., 2022). Recent advancements in video foundation models (Yu et al., 2023; Guo et al., 2023; Wang et al., 2023a; Yang et al., 2024e) have led efforts like VideoSwap (Gu et al., 2023) to employ temporal priors for customized motion transfer or motion editing (Mou et al., 2025). Yet, current video foundation models are limited to fixed views and struggle with non-grid human motions. Additionally, these editing methods require tuning parameters, which poses a challenge for real-time video editing applications. In contrast, our VideoGrain method requires no parameter tuning, enabling zero-shot, multi-grained video editing.

## 3 METHOD

### 3.1 MOTIVATION

To investigate why previous methods failed in instance-level video editing (see Fig. 2), we begin with a basic analysis of the self-attention and cross-attention features within the diffusion model.

As shown in Fig. 3 (b), we apply K-Means clustering to the per-frame self-attention features during DDIM Inversion. Although the clustering captures a clear semantic layout, it fails to distinguish between distinct instances (e.g., "left man" and "right man"). Increasing the number of clusters leads to finer segmentation at the part level but does not resolve this issue, indicating that feature homogeneity across instances limits the diffusion model's effectiveness in multi-grained video editing.

Next, we attempt to edit the same class of two men into different instances using SDEdit (Meng et al., 2022). However, Fig. 3 (d) shows that the weights for "Iron Man" and "Spiderman" overlap on the left man, and "blossoms" weight leaks onto the right man, resulting in the failed edit in (c).

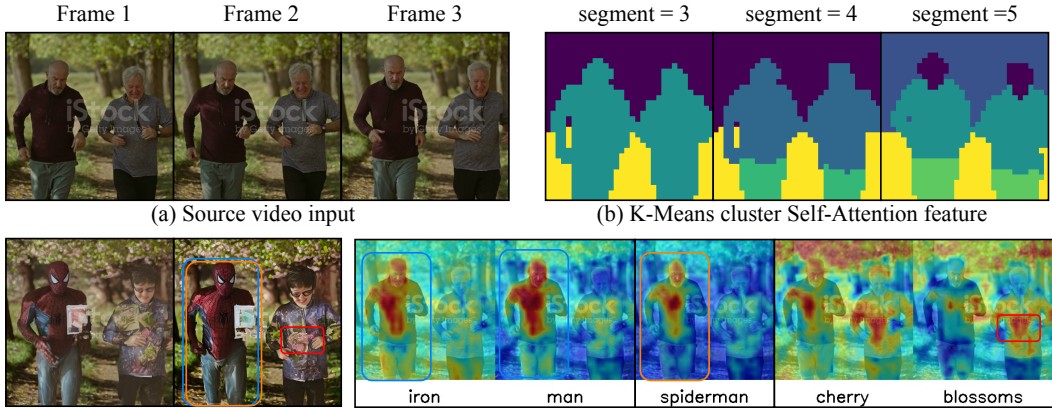

Frame 1    Frame 2    Frame 3          segment = 3    segment = 4    segment =5

(a) Source video input          (b) K-Means cluster Self-Attention feature

(c) Instance-level failed case    (d) Cross-Attention Map: "*An Iron Man and a Spiderman are jogging under cherry blossoms*"

**Figure 3: Analysis of why the diffusion model failed in instance-level video editing**. Our goal is to edit left man into "Iron Man," right man into "Spiderman," and trees into "cherry blossoms." In (b), we apply K-Means on self-attention, and in (d), we visualize the 32x32 cross-attention map.

Thus, for effective multi-grained editing, we pose the following question: *Can we modulate attention to ensure that each local edit's attention weights are accurately distributed in the intended regions?*

To answer this, we propose VideoGrain with two key designs: (1) Modulate cross-attention to induce textual features to congregate in corresponding spatial-disentangled regions, thereby enabling text-to-region control. (2) Modulate self-attention across the spatial-temporal axis to enhance intra-region focus and reduce inter-region interference, avoiding feature coupling within diffusion model.

## 3.2   PROBLEM FORMULATION

The purpose of this work is to perform multi-grained video editing across multiple regions based on the given prompts. This involves three hierarchical levels:

**(1) Class-level editing**: Editing objects within the same class. (e.g., changing two men to "Spiderman," where both belong to the human class, as seen in Fig. 2 second column)

**(2) Instance-level editing**: Editing each individual instance to distinct object. (e.g., editing left man to "Spiderman," right man to "Polar Bear," as shown in Fig. 2 third column).

**(3) Part-level editing**: Applying part-level edit to specific elements of individual instances. (e.g., adding "sunglasses "when editing the right man to "Polar Bear" in Fig. 2 fourth column).

Given a source video $\mathbf{V} \in \mathbb{R}^{N \times 3 \times H \times W}$, where $N$ is the number of frames, our goal is to obtain an edited video $\mathbf{V}'$ based on specified edits. We aim to improve multi-grained control in video editing by conditioning on each region's location and its text prompt. More formally, we optimize a video editing model $f(\tau_g, (\tau_1, m_1), \ldots, (\tau_k, m_k))$, where $\tau_g$ is a global prompt, and $(\tau_k, m_k)$ are the $k_{th}$ region's prompt and corresponding location.

## 3.3   OVERALL FRAMEWORK

The proposed zero-shot multi-grained video editing pipeline is illustrated in Fig. 4 top. Initially, to retain high fidelity, we perform DDIM Inversion (Song et al., 2021) over the clean latent $\boldsymbol{x}_0$ to get the noisy latent $\boldsymbol{x}_t$. After the inversion process, we cluster the self-attention features to get the semantic layout as in Fig. 3 (b). Since self-attention features alone cannot distinguish between individual instances, we further employ SAM-Track (Cheng et al., 2023) to segment each instance. Finally, in the denoising process, we introduce ST-Layout Attn to modulate cross- and self-attention for text-to-region control and keep feature separation between regions, as detailed in Sec. 3.4.

Different from one global text prompt control of all frames, VideoGrain allows paired instance- or part-level prompts and their locations to be specified in the denoising process. Our method is also versatile to ControlNet condition $e$, which can be depth or pose maps to provide structure conditions.

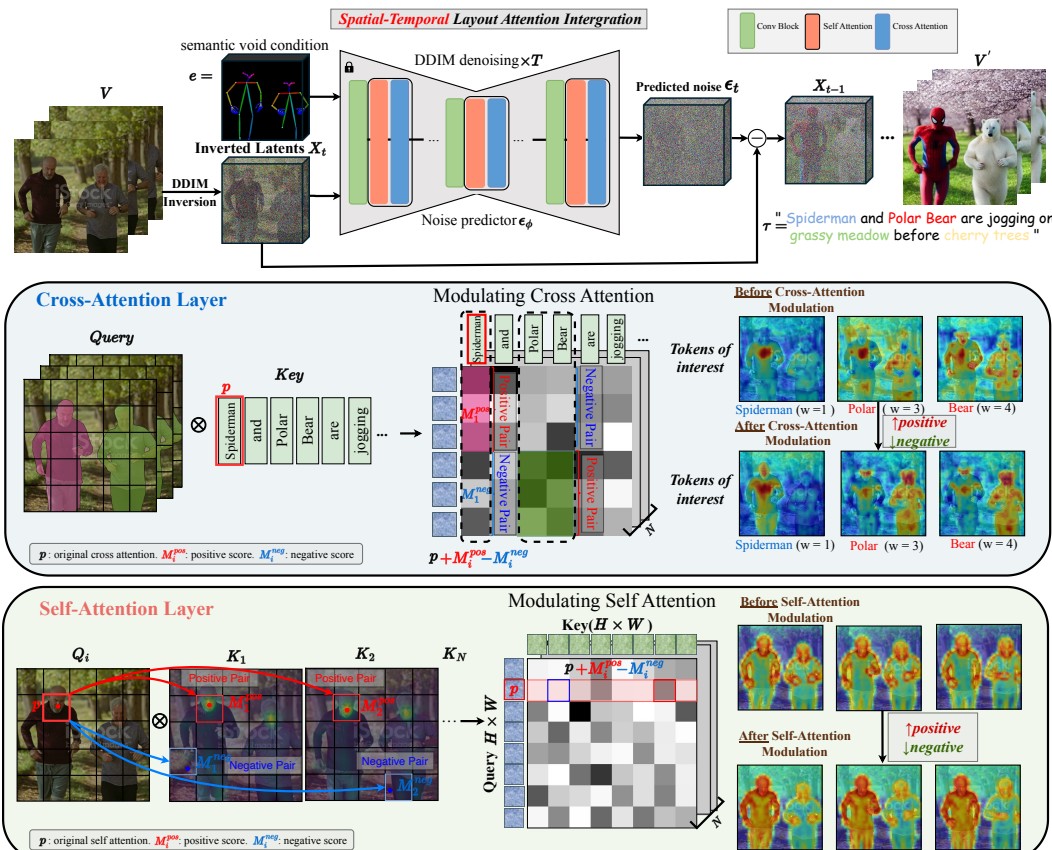

**Figure 4:** VideoGrain pipeline. (1) we integrate ST-Layout Attn into the frozen SD for multi-grained editing, where we modulate self- and cross-attention in a unified manner. (2) In cross-attention, we view each local prompt and its location as positive pairs, while the prompt and outside-location areas are negative pairs, enabling text-to-region control. (3) In self-attention, we enhance positive awareness within intra-regions and restrict negative interactions between inter-regions across frames, making each query only attend to the target region and keep feature separation. In the bottom two figures, $p$ denotes original attention score and $w, i$ denotes the word and frame index.

## 3.4 SPATIAL-TEMPORAL LAYOUT-GUIDED ATTENTION

Based on the observation in Sec.3.1, cross-attention weight distribution adheres to the edit result. Meanwhile, self-attention is also crucial to generate temporal consistent video. However, the pixels in one region may attend to outside or similar regions, which poses an obstacle for multi-grained video editing. Therefore, we need to modulate both self- and cross-attention to make each pixel or local prompt only focus on the correct region.

To achieve this goal, we modulate both cross- and self-attention mechanisms via a unified increase positive and decrease negative manner. Specifically, for the $i_{th}$ frame of the query feature, we modulate the query-key $QK^\top$ condition map as follows:

$$
\begin{aligned}
A_i^{\text{self/cross}} &= \text{softmax}\left(\frac{QK^\top + \lambda M^{\text{self/cross}}}{\sqrt{d}}\right), \\
M^{\text{self/cross}} &= R_i \odot M_i^{\text{pos}} - (1 - R_i) \odot M_i^{\text{neg}},
\end{aligned}
\tag{1}
$$

where $R_i \in \mathbb{R}^{|\text{queries}| \times |\text{keys}|}$ indicates the query-key pair condition map at frame $i$, manipulating whether to increase or decrease the attention score for a particular pair. And $\lambda = \xi(t) \cdot (1 - S_i)$ is a regularization term. We follow the conclusion from (Kim et al., 2023), the $\xi(t)$ controls the modulation intensity across time-steps, allowing for gradual refinement of shape and appearance details. The latter is a size regulation term, making smaller region $m_k$ subjected to larger modulation, enabling dynamic attention weight adjustments to layout size variations.

**Modulate Cross-Attention for Text-to-Region Control.** In the cross-attention layer, the textual feature serves as key and value, and interacts with the query feature from the video latent. Since each instance's appearance and location are closely related to the cross-attention weight distribution, we aim to encourage each instance's textual features to congregate in the corresponding location.

As shown in Fig. 4 mid, given the layout condition $(\tau_k, m_k)$. For example, for $\tau_1 = $ Spiderman, within the query-key cross-attention map, we can manually specify that the portion of the query feature corresponding to $m_1$ is positive, while all the remaining parts are designated as negative. Therefore, for each frame $i$, we can set the modulation value in cross attention layer as:

$$
\begin{aligned}
M_i^{\text{pos}} &= \max(QK^\top) - QK^\top, \\
M_i^{\text{neg}} &= QK^\top - \min(QK^\top),
\end{aligned}
\tag{2}
$$

$$
R_i^{\text{cross}}[x,y] = \left\{ \begin{array}{ll} m_{i,k}, & \text{if } y \in \tau_k \\ 0, & \text{otherwise} \end{array} \right. ,
\tag{3}
$$

where $x$ and $y$ are the query and key indices, and $R_i^{\text{cross}}$ is the query-key condition map in the cross attention layer. We regularize this condition map by initially broadcasting each region's mask $m_{i,k}$ to its corresponding text key embedding $K_{\tau_k}$, resulting in a condition map $R_i^{\text{cross}} \in \mathbb{R}^{(H \times W) \times L}$. Each sub-region intensity then adjusts gradually in the generation process. We set $M_i^{\text{pos/neg}}$ based on the gap between max/min values and the original scores, to keep modulated values within the original range. Our modulation is applied to all frames to achieve spatial-temporal region control.

As shown in Fig. 4 (mid right), after adding positive and subtracting negative values, the original cross-attn weight of "Spiderman" (e.g., $p$) is amplified and focused on the left man. While the distract weight of "polar" "bear" become concentrated on the right man. These indicate our modulation redistributes each prompt's weight align with target areas, enabling precise text-to-region control.

**Modulate Self-Attention to Keep Feature Separation.** To adapt the T2I model for T2V editing, we treat the full video as "a larger picture," replacing spatial attention with spatial-temporal self-attention while retaining the pretrained weights. This enhances cross-frame interaction and provides a broader visual context. However, naive self-attention can cause regions to attend to irrelevant or similar areas (e.g., Fig. 4 bottom, before modulation query $p$ attend to two-man), which leads to mixed texture. To address this, we need to strengthen positive focus within the same region and restrict negative interactions between different regions.

As shown in Fig. 4 (bottom left), the maximum cross-frame diffusion feature indicates the strongest response among tokens within the same region. Note that DIFT (Tang et al., 2023) uses this to match different images, while we focus on cross-frame correspondences and intra-region attention modulation in the generation process. Nevertheless, negative inter-region correspondence is equally crucial for decoupling feature mixing. Beyond DIFT, we find that the minimum cross-frame diffusion feature similarity effectively captures the relations between tokens across different regions. Therefore, we define the spatial-temporal positive/negative values as:

$$
\begin{aligned}
M_i^{\text{pos}} &= \max(Q_i[K_1, \cdots, K_n]^\top) - Q_i[K_1, \cdots, K_n]^\top, \\
M_i^{\text{neg}} &= Q_i[K_1, \cdots, K_n]^\top - \min(Q_i[K_1, \cdots, K_n]^\top).
\end{aligned}
\tag{4}
$$

To ensure each patch attends to intra-regions feature while avoiding interaction in inter-regions feature. We define the spatial-temporal query-key condition map:

$$
R_i^{\text{self}}[x,y] = \left\{ \begin{array}{l} 0, \forall j \in [1:N], \text{if } m_{i,k}[x] \neq m_{j,k}[y] \\ 1, \text{otherwise} \end{array} \right. .
\tag{5}
$$

For frame indices $i$ and $j$, the value is zero when tokens belong to different instances across frames.

As shown in the right part of Fig. 4 bottom, after applying our self-attention modulation, the query feature from the left man's nose (e.g., $p$) attends only to the left instance, avoiding distraction to the right instance. This demonstrates that our self-attention modulation breaks the diffusion model's class-level feature correspondence, ensuring feature separation at the instance level.

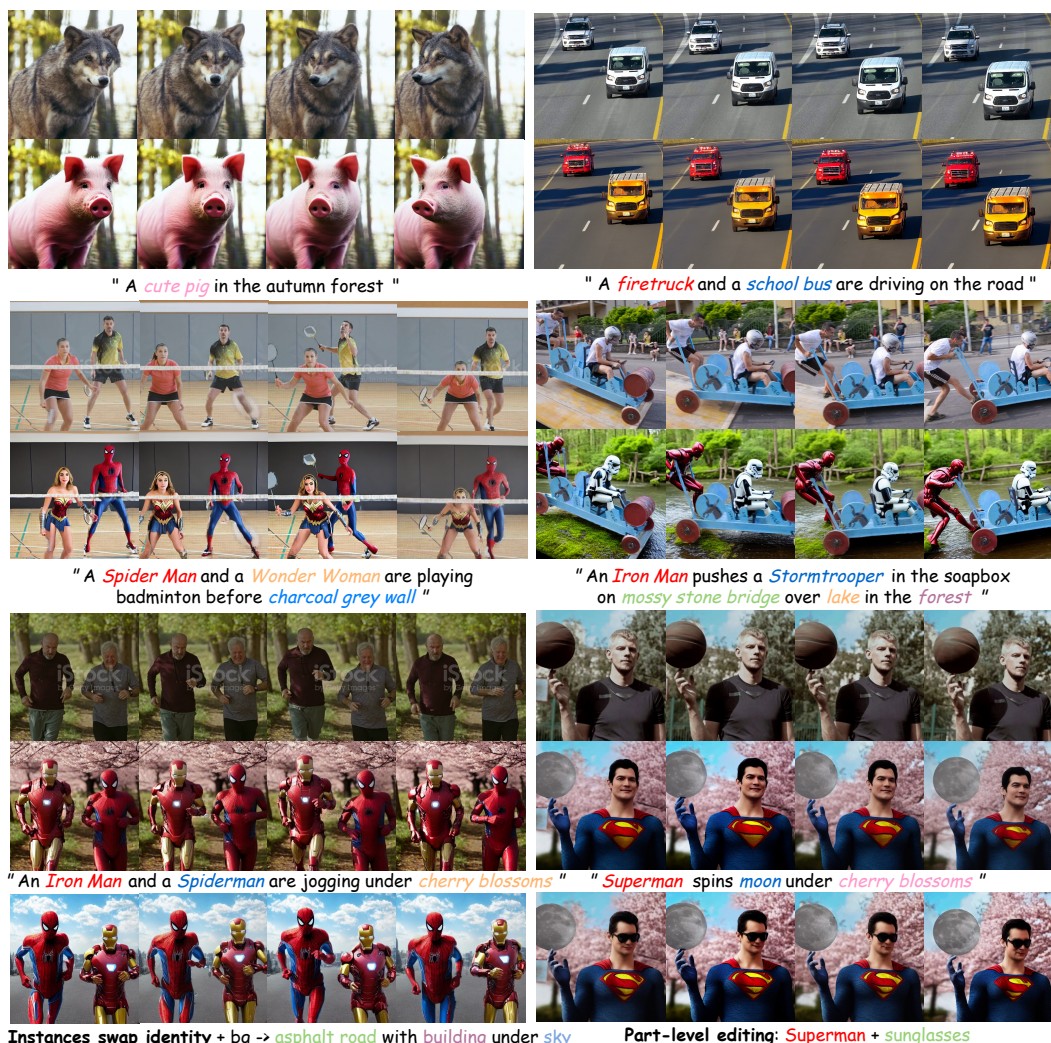

**Figure 5:** Qualitative results. VideoGrain achieves multi-grained video editing, including class-level, instance-level, and part-level. We refer the reader to our project page for full-video results.

# 4 EXPERIMENTS

## 4.1 EXPERIMENTAL SETTINGS

In the experiment, we adopt the pretrained Stable Diffusion v1.5 as the base model, using 50 steps of DDIM inversion and denoising. Our VideoGrain operates in a zero-shot manner, requiring no additional parameter tuning. To enhance memory efficiency, we re-engineer slice attention within our ST Layout Attn. ST Layout Attn is applied during the first 15 denoising steps. We set $\xi(t) = 0.3 \cdot t^5$ for self-attention and $\xi(t) = t^5$ for cross-attention, where the timestep $t \in [0, 1]$ is normalized. All The experiments are conducted on an NVIDIA A40 GPU. We evaluate our VideoGrain using a dataset of 76 video-text pairs, including videos from DAVIS (Perazzi et al., 2016), TGVE[1], and the Internet[2] , with 16-32 frames per video. Four automatic metrics are employed for evaluation: CLIP-T, CLIP-F, Warp-Err, and Q-edit, following (Wu et al., 2022; Cong et al., 2023). All metrics are scaled by 100 for clarity. For baselines, we compare against T2I-based methods, including FateZero (Qi et al., 2023), ControlVideo (Zhang et al., 2023b), TokenFlow (Geyer et al., 2023), GroundVideo (Jeong & Ye, 2023) and T2V-based DMT (Yatim et al., 2024). To ensure temporal consistency, we employ FLATTEN (Cong et al., 2023) and PnP (Tumanyan et al., 2023). For fairness, all T2I baselines are equipped with the same ControlNet conditions.

---

[1] https://sites.google.com/view/loveucvpr23/track4
[2] https://www.istockphoto.com/ and https://www.pexels.com/

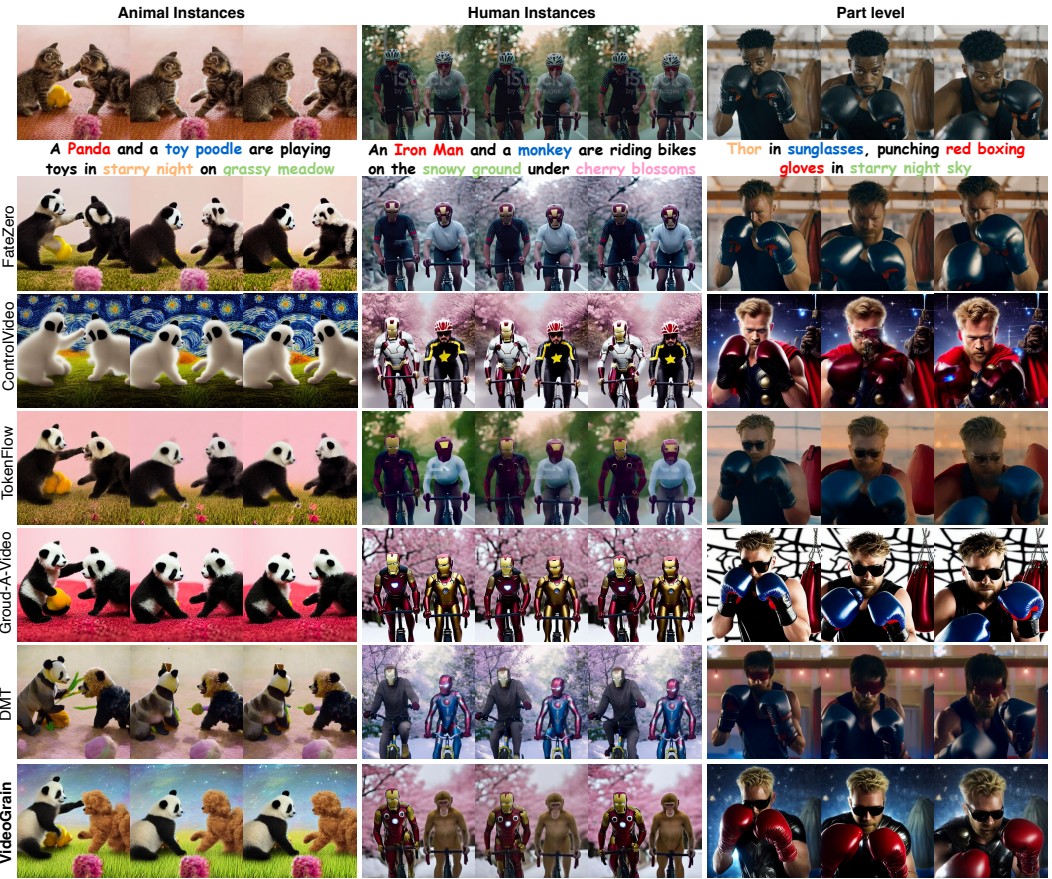

**Figure 6:** Qualitative comparisons. We refer the reader to our project page for detailed assessment.

## 4.2 RESULTS

We evaluate VideoGrain on videos covering class-level, instance-level, and part-level edits. Our method demonstrates versatility in handling animals, such as transforming a "wolf" into a "pig" (Fig. 5, top left). For instance-level editing, we can modify vehicles separately (e.g., transforming an "SUV" into a "firetruck" and a "van" into a "school bus") in Fig. 5, top right. VideoGrain excels at editing multiple instances in complex, occluded scenes, like "Spider-Man and Wonder Woman playing badminton" (Fig. 5, middle left). Previous methods often struggle with such non-rigid motion. In addition, our method is capable of multi-region editing, where both foreground and background are edited, as shown in the soap-box scene, where the background changes to "a mossy stone bridge over a lake in the forest" (Fig. 5, middle right). Thanks to precise attention weight distribution, we can swap identities seamlessly, such as in the jogging scene, where "Iron Man" and "Spider-Man" swap identities (Fig. 5, bottom left). For part-level edits, VideoGrain excels in adjusting a character to wear a Superman suit while keeping sunglasses intact (Fig. 5, bottom right). Overall, for multi-grained editing, our VideoGrain demonstrates outstanding performance.

## 4.3 QUALITATIVE AND QUANTITATIVE COMPARISONS

**Qualitative Comparison.** Figure 6 shows a comparison between VideoGrain and baseline methods, including T2I-based and T2V-based approaches, for instance-level and part-level editing. For fairness, all T2I-based methods use ControlNet conditioning. **(1) Animal instances**: In the left column, T2I-based methods like FateZero, ControlVideo, and TokenFlow edit both cats into pandas due to same-class feature coupling in diffusion models, failing to perform separate edits. DMT, even with video generation priors, still blends the panda and toy poodle features. In contrast, VideoGrain successfully edits one into a panda and the other into a toy poodle. **(2) Human instances**: In the middle column, baselines struggle with same-class feature coupling, partially editing both men into

| Method | Automatic Metric | | | | Human Evaluation | | |
|---|---|---|---|---|---|---|---|
| | CLIP-F ↑ | CLIP-T ↑ | Warp-Err ↓ | Q-edit ↑ | Edit-Acc ↑ | Temp-Con ↑ | Overall ↑ |
| FateZero | 95.75 | 33.78 | 3.08 | 10.96 | 59.8 | 78.6 | 59.6 |
| ControlVideo | 97.71 | 34.41 | 4.73 | 7.27 | 53.2 | 50.0 | 43.6 |
| TokenFlow | 96.48 | 34.59 | 2.82 | 12.28 | 45.4 | 50.4 | 39.8 |
| Ground-A-Video | 95.17 | 35.09 | 4.43 | 7.92 | 69.0 | 72.0 | 63.2 |
| DMT | 96.34 | 34.09 | 2.05 | 16.63 | 58.7 | 79.4 | 64.5 |
| **VideoGrain(ours)** | **98.63** | **36.56** | **1.42** | **25.75** | **88.4** | **85.0** | **83.0** |

**Table 1:** Quantitative comparison of automatic metrics and human evaluation. The best results are **bolded**.

Iron Man. DMT and Ground-A-Video also fail to follow user intent, incorrectly editing the left and right instances. VideoGrain, however, correctly transforms the right man into a monkey, breaking the human-class limitation. **(3) Part-level editing**: In the third column, VideoGrain manages part-level edits, such as sunglasses and boxing gloves. ControlVideo edits the gloves but struggles with sunglasses and motion consistency. TokenFlow and DMT edit the sunglasses but fail to modify the gloves or background. In comparison, VideoGrain achieves both instance-level and part-level edits, significantly outperforming previous methods.

**Quantitative Comparison.** We compare the performance of different methods using both automatic metrics and human evaluation. **CLIP-T** calculates the average cosine similarity between the input prompt and all video frames, while **CLIP-F** measures the average cosine similarity between consecutive frames. Additionally, **Warp-Err** captures pixel-level differences by warping the edited video frames according to the optical flow of the source video, extracted using RAFT-Large (Teed & Deng, 2020). To provide a more comprehensive measure of video editing quality, we follow (Cong et al., 2023) and use **Q-edit**, defined as CLIP-T/Warp-Err. For clarity, we scale all automatic metrics by 100. In terms of human evaluation, we assess three key aspects: **Edit-Accuracy** (whether each local edit is accurately applied), **Temporal Consistency** (evaluated by participants for coherence between video frames), and **Overall Edit Quality**. We invited 20 participants to rate 76 video-text pairs on a scale of 20 to 100 across these three criteria, following (Jeong & Ye, 2023). As demonstrated in Table 1, VideoGrain consistently outperforms both T2I- and T2V-based methods. This is primarily due to ST-Layout Attn's precise text-to-region control and maintaining feature separation between regions. As a result, our method achieves significantly higher CLIP-T and Edit-Accuracy scores compared to other baselines. The improved Warp-Err and Temporal Consistency metrics further indicate that VideoGrain delivers temporally coherent video edits.

**Efficiency Comparison.** To evaluate efficiency, we compared baselines with VideoGrain on a single A6000 GPU for editing 16 video frames. The metrics include editing time (time taken to perform one edit) and both GPU and CPU memory usage. From Tab. 2, it is clear our method achieves the fastest editing time with the lowest memory usage, indicating its computational efficiency.

| | Time(min) ↓ | Memory (GB) ↓ | RAM (GB) ↓ |
|---|---|---|---|
| FateZero | 8.68 | 27.35 | 144.22 |
| ControlVideo | 4.41 | 16.15 | 7.03 |
| TokenFlow | 4.56 | 17.84 | 5.35 |
| Ground-A-Video | 5.81 | 17.31 | 9.96 |
| DMT | 5.79 | 27.88 | 8.12 |
| **VideoGrain** | **3.83** | **15.94** | **4.42** |

**Table 2:** Efficiency comparison.

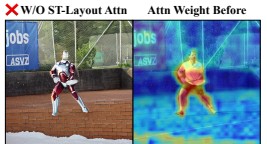 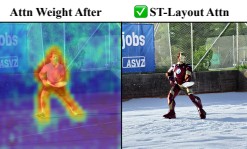

**Figure 7:** Attention weight distribution.

### 4.4 ABLATION STUDY

To assess the contributions of different components in our proposed ST-Layout Attn, we first evaluate whether our attention can achieve attention weight distribution, then decouple the self-attention modulation and cross-attention modulation to evaluate their individual effectiveness.

**Attention Weight Distribution.** We evaluate the impact of ST-Layout Attn on attention weight distribution. As shown in Fig. 7, the target prompt is "An Iron Man is playing tennis on a snow court." We visualize the cross-attention map for "man" to assess weight distribution. Without ST-Layout Attn, feature mixing occurs, with "snow" weight spilling onto "Iron Man." With ST-Layout Attn, the man's weight is correctly distributed. This is because we enhance positive pair scores and suppress negative pairs in both cross- and self-attention. This enables precise, separate edits for "Iron Man" and "snow." Additional visualizations are in the Appendix.

left man → *Iron Man*, right man → *Spider man*, ground, trees → *frosty yellow leaves*

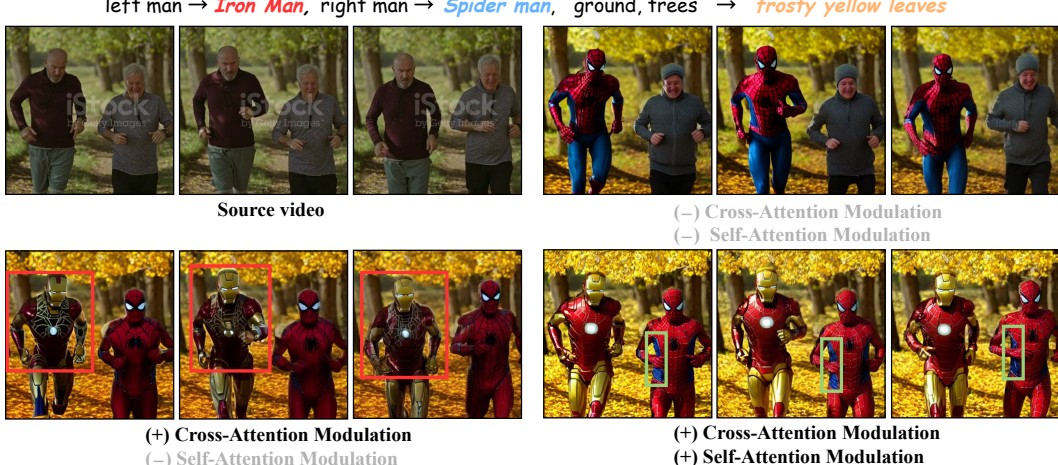

**Figure 8:** Ablation of cross- and self-modulation in ST-Layout Attn.

| Method | CLIP-F ↑ | CLIP-T ↑ | Warp-Err ↓ | $Q_{edit}$ ↑ |
|---|---|---|---|---|
| Baseline | 95.21 | 33.59 | 3.86 | 8.70 |
| Baseline + Cross Modulation | 96.28 | 36.09 | 2.53 | 14.26 |
| **Baseline + Cross Modulation + Self Modulation** | **98.63** | **36.56** | **1.42** | **25.75** |

**Table 3:** Quantitative ablation of cross- and self-modulation in ST-Layout Attn.

**Cross-Attention Modulation.** In Fig. 8 and Tab. 3, we illustrate video editing results under different set up: (1) Baseline (2) Baseline + Cross-Attn Modulation (3) Baseline + Cross-Attn Modulation + Self-Attn Modulation. As shown in Fig. 8 top right, direct editing fails to discriminate between the left and right instances, leading to incorrect (left) or no edits(right). However, when equipped with cross-attention modulation, we achieve accurate text-to-region control, thereby editing left man to "Iron Man" and right man to "Spiderman" separately. The quantitative results in Tab. 3 indicate that with cross-attention modulation (second row), CLIP-T increases by 7.4%, and Q-edit increases by 63.9%. This demonstrates the effectiveness of our cross-attention modulation.

**Self-Attention Modulation.** However, modulating only cross-attention still leads to structure distortions, such as the spider web appearing on the left man. This is caused by the coupling of same class-level features (e.g., human). When using our self-attention modulation, the feature mixing is significantly reduced, and the left man retains unique object features. This is achieved by decreasing the negative pair scores between different instances, while increasing positive scores within the same instance. As a result, more part-level details, such as the distinctive blue sides, are generated in the optimized areas. The quantitative decrease in Warp-Err by 43.9% and increase in Q-edit by 80.6% in Tab. 3 further prove the effectiveness of self-attention modulation.

## 5 CONCLUSION

In this paper, we aim to solve the problem of multi-grained video editing, which includes both class-level, instance-level and part-level video editing. To the best of our knowledge, this is the first attempt at this task. In this task, we find that the key problem is that the diffusion model views different instances as same-class features and direct global editing will mix different local regions. To wrestle with these problems, we propose VideoGrain to modulate spatial-temporal cross- and self-attention for text-to-region control while keeping feature separation between regions. In cross-attention, we enhance each local prompt's focus on its corresponding spatial-disentangled region while suppressing attention to irrelevant areas, thereby enabling text-to-region control. In self-attention, we increase intra-region awareness and reduce inter-region interactions to keep feature separation between regions. Extensive experiments demonstrate that our VideoGrain surpasses previous video editing methods on both class-level, instance-level, and part-level video editing.

## 6 ETHICS STATEMENT

This project aims to solve multi-grained video editing. However, the potential misuse of this technology, such as the creation of deceptive videos by altering identities, poses a risk. Strategies like incorporating invisible watermarking could be explored to ensure videos are not used maliciously.

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

Different from multi-modal learning (Yang et al., 2024b;c;d; Jia et al., 2024), controllable generation (Ma et al., 2024b;a; Lu et al., 2024; Zhou et al., 2024) or video editing (Yang et al., 2024a; Ma et al., 2023) requires explicit control signals. Multi-grained editing further relies on additional layout conditions to edit in the class, instance, or part level. Therefore, in the appendix, we first evaluate the SAM-Track masks' impact in Section A, then validate whether our method can work without SAM-Track masks in Section B. Continually, we show that our method can solely edit specific subjects in Section C and part-level modification example in Section D. We also evaluate our ST-Layout Attn's temporal focus in Section E and ControlNet's effect in Section F.

## A  EVALUATE SAM-TRACK MASKS' IMPACT

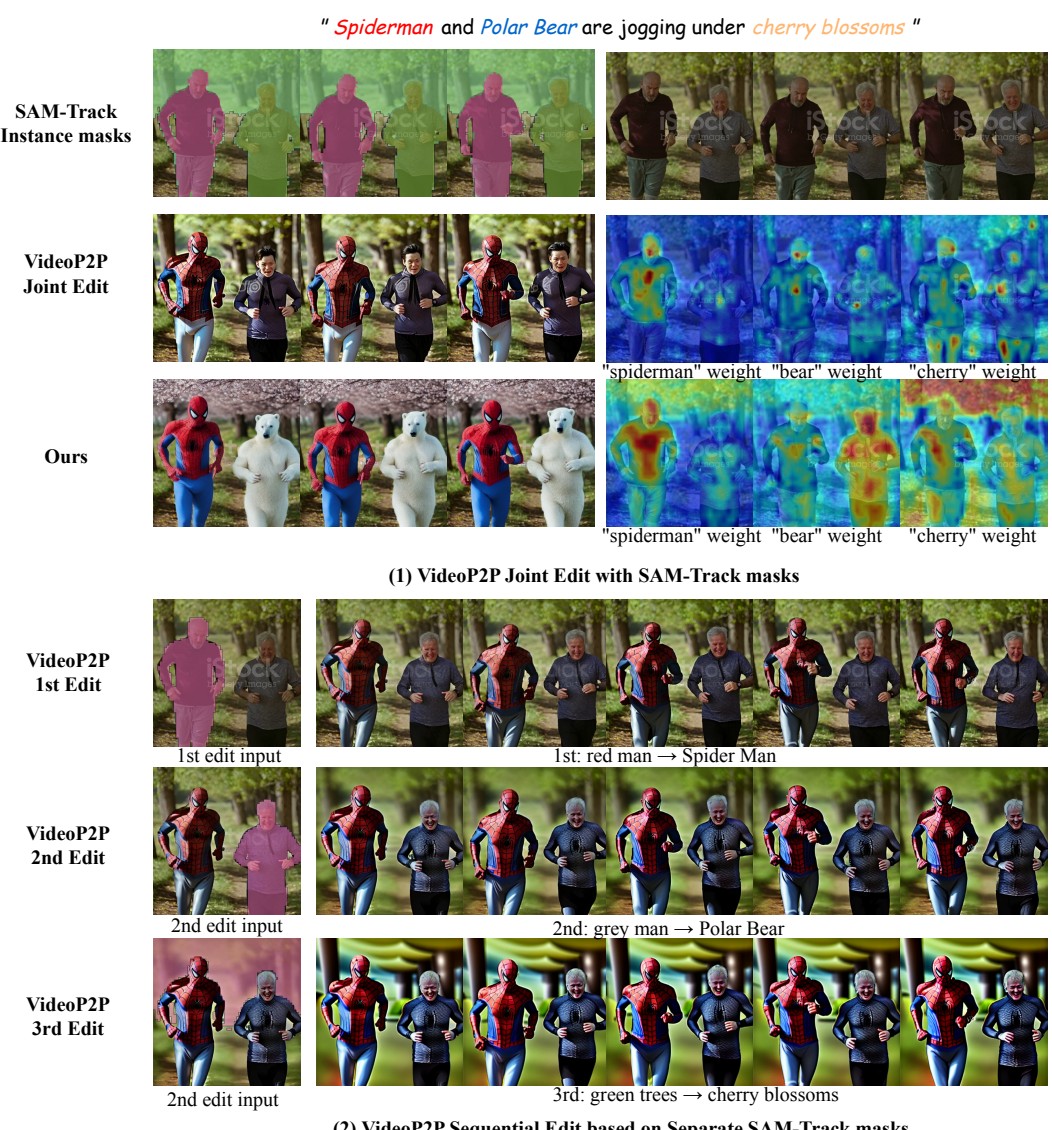

**Figure 9:** VideoP2P joint and sequential edit with SAM-Track masks

To evaluate the impact of using SAM-Track (Cheng et al., 2023) for instance segmentation, we compare our VideoGrain against VideoP2P (Liu et al., 2024), which is equipped with SAM-Track instance masks. The instance masks replace cross-attention masks during editing. A 16-frame one-shot tuning is performed, and ControlNet conditioning Zhang et al. (2023a) is added for fairness.

Two experiments are tested: (1) jointly editing multiple areas in a single denoising process and (2) sequentially editing three areas by inputting separate masks.

Results show that joint editing (Fig. 9(1)) modifies only left man into "Spiderman," leaving other areas unchanged due to inaccurate cross-attn weight distribution. Sequential editing (Fig. 9(2)) succeeds initially but fails later due to error accumulation in denoising, resulting in blurred details.

*" Spiderman and Polar Bear are jogging under cherry blossoms "*

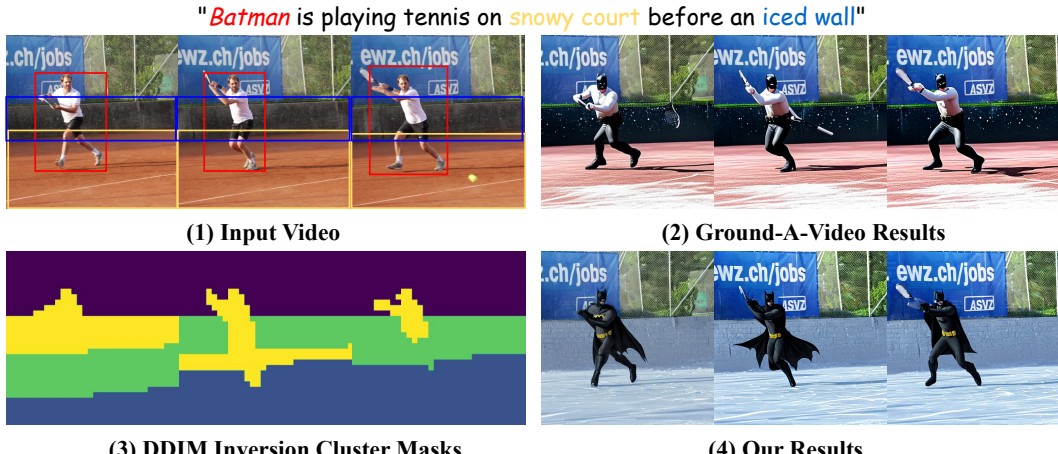

Ground-A-Video Input          Ground-A-Video Result

**Figure 10:** Ground-A-Video joint edit with instance information

Additionally, as shown in figure above 10, also in Figs 2 and 6, Ground-A-Video (Jeong & Ye, 2023) struggles with multi-grained video editing tasks, even with instance-level grounding information (e.g., text-to-bounding box), which is comparable to SAM-Track's masks.

These comparisons indicate that while SAM-Track provides layout guidance, it does not guarantee successful edits. In contrast, our method enables zero-shot multi-grained editing, which was not achievable by any previous methods, even when providing existing SOTA with SAM-Track masks.

## B   VIDEOGRAIN CAN WORK WITHOUT SAM-TRACK MASKS

*"Batman is playing tennis on snowy court before an iced wall"*

**(1) Input Video**          **(2) Ground-A-Video Results**

**(3) DDIM Inversion Cluster Masks**          **(4) Our Results**

**Figure 11:** Our method without additional SAM-Track masks

Our method is not strictly dependent on SAM-Track masks. As shown in Fig.11(3), we can cluster DDIM inversion self-attention features to get inaccurate coarse layouts. Our method still achieves high-quality multi-area editing results (4). In contrast, even with precise groundings (converted from SAM-Track masks in (1)), Ground-A-Video fails to edit all three regions. These comparisons indicate that our method does not rely on SAM-Track segmentation. Instead, it works effectively only using the self-attention feature inside the diffusion model, even without accurate layout guidance.

## C   SOLELY EDIT ON SPECIFIC SUBJECTS, WITHOUT BACKGROUND CHANGED

Our method is designed for multi-area editing and can naturally perform background-preserved subject editing, as it treats multi-area editing as selecting regions restricted to the foreground. As

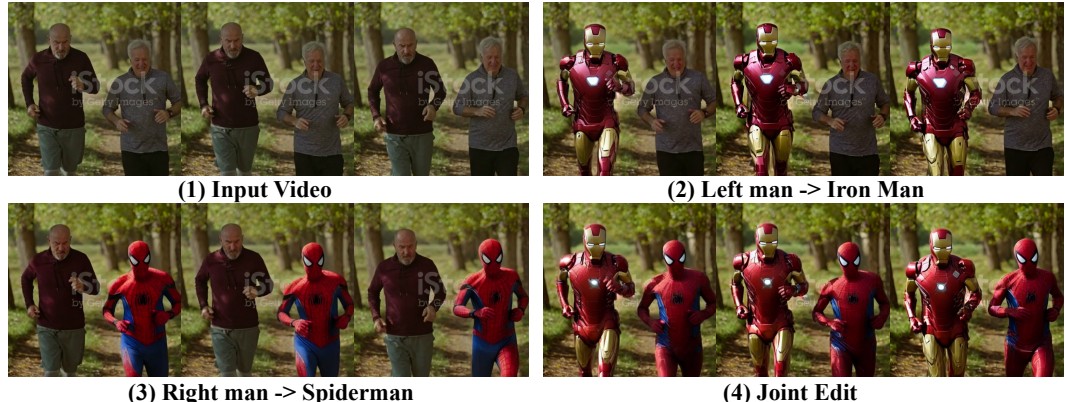

| | |
|---|---|
| **(1) Input Video** | **(2) Left man -> Iron Man** |
| **(3) Right man -> Spiderman** | **(4) Joint Edit** |

**Figure 12:** Soely edit on specific subjects, without background changed

shown in Fig 12, our method can separately edit the "left man" and "right man" or jointly edit both subjects while keeping the background unchanged.

## D  PART-LEVEL MODIFICATION EXAMPLES

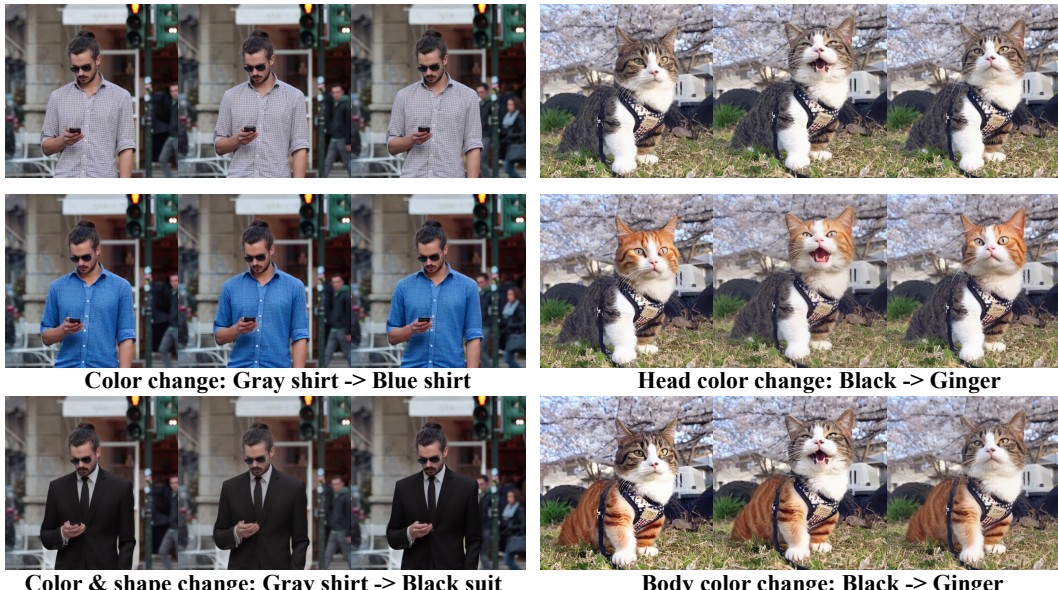

| | |
|---|---|
| **Color change: Gray shirt -> Blue shirt** | **Head color change: Black -> Ginger** |
| **Color & shape change: Gray shirt -> Black suit** | **Body color change: Black -> Ginger** |

**Figure 13:** Part-level modifications on humans and animals

Our part-level editing supports not only adding objects but also part-level attribute modifications. In the human case (Fig. 13 left), our method changes the color of a gray shirt to blue (second row) and edits a half-sleeve shirt into a black suit (third row), showcasing part-level attribute and structure editing. Similarly, in the animal case, our method can change a cat's head or body color from black to ginger while preserving the belt's color, demonstrating precise part-level modifications.

## E  TEMPORAL FOCUS OF ST-LAYOUT ATTN

Our ST-Layout Attn is designed as a full-frame approach to ensure inter-frame consistency. As shown in Fig. 14, per-frame ST-Layout Attn causes feature coupling on Iron Man, while the sparse-causal method results in flickering and misses Spider Man's blue details due to their limited receptive fields for positive/negative value selection across different layouts. In contrast, our ST-Layout Attn

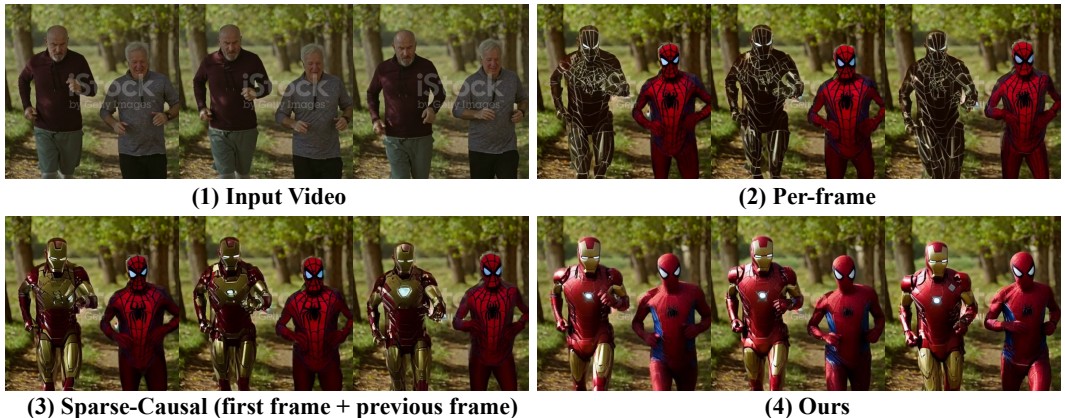

**Figure 14:** Temporal Focus of ST-Layout Attn

effectively preserves texture details and prevents flickering, achieving temporal consistent and layout unified multi-grained video editing.

## F    CONTROLNET ABLATION

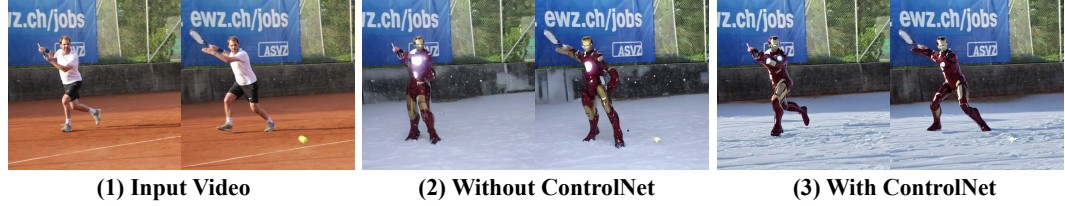

**Figure 15:** ControlNet ablation

Our method utilizes ControlNet depth/pose conditioning in certain complex motion cases to provide necessary structural guidance. As shown in Fig. 15, even without ControlNet, our method can still achieve simultaneous multi-region editing. However, in such cases, there may be some structural inconsistencies between the edit object and source object due to the lack of explicit structure guidance.

## G    MORE GENERAL OBJECTS AND SHAPE EDITING

## H    MORE VISUALIZATION

## I    LATENT BLEND

To preserve areas not intended for editing (i.e., $\tau_3$ in $\Delta_\tau = \{\tau_1 \rightarrow \tau_{1'}, \tau_2 \rightarrow \tau_{2'}, \tau_3 \rightarrow \tau_3, \cdots \}$), we employ Latent Blend (Avrahami et al., 2022; 2023), which leverages masks to direct the model focus on areas requiring editing while keeping the background region identical to the source video.

For each frame $i$ in the video, we first merge each attribute mask to form the global foreground mask $M_i$ by applying the logical OR operation across all layouts masks $m_{i,k} = [m_{i,1}, m_{i,2}, \cdots, m_{i,k}]$ :

$$M_i = m_{i,1} \vee m_{i,2} \vee \cdots \vee m_{i,k}. \qquad (6)$$

We aggregate the masks $M_i$ from all frames to obtain a combined mask $M$, and then blend the latent states $z_t$ at each timestep $t$ during the denoising process as follows:

$$z_t = (1 - \mathcal{M}) \cdot \tilde{z}_t + \mathcal{M} \cdot z_t, \qquad (7)$$

where $\tilde{z}_t$ indicates the latent feature in the DDIM inversion process and $z_t$ is corresponding latent feature during the DDIM denoising process.

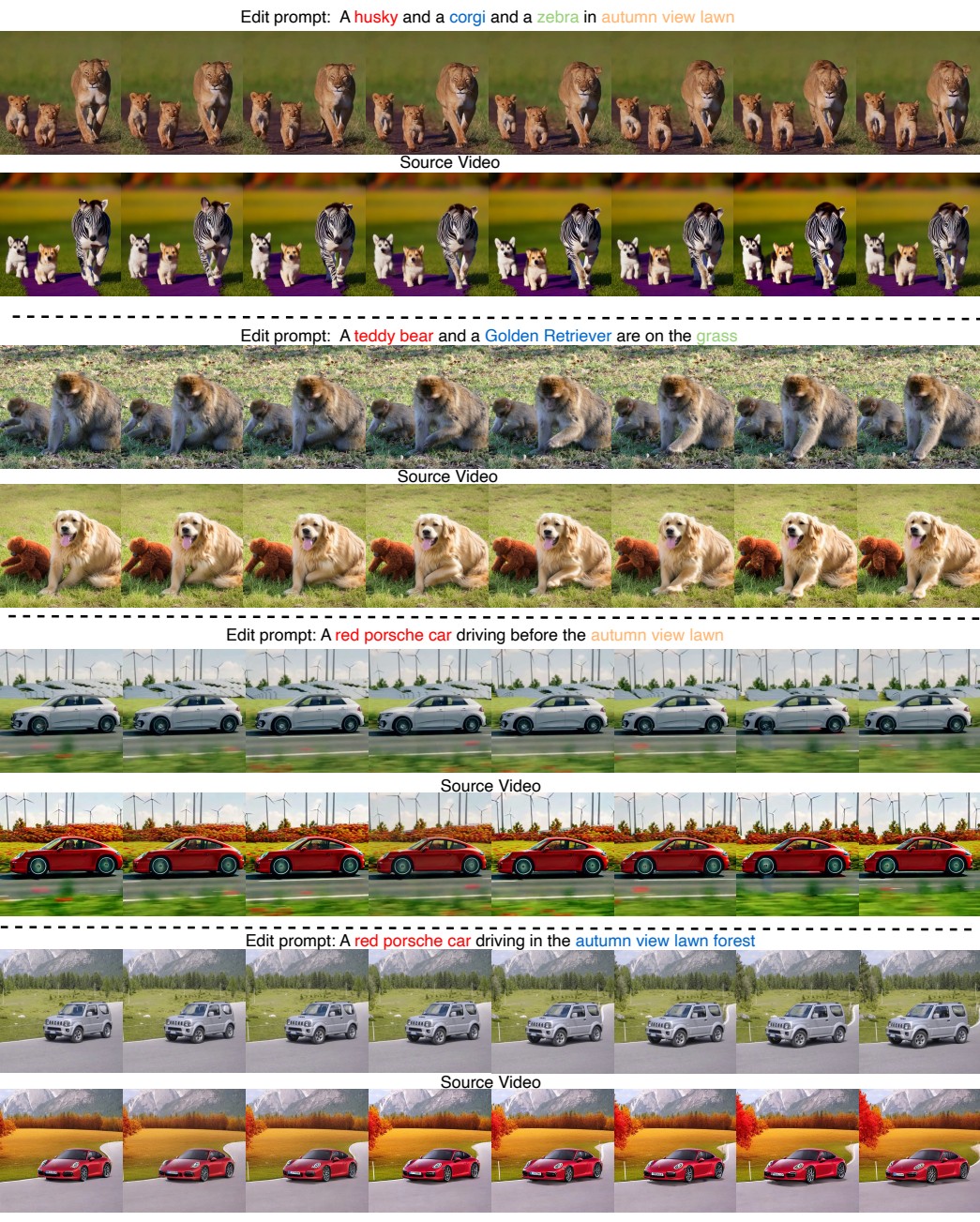

**Figure 16:** More general objects instance editing (animals) and shape editing (cars) results.

The key behind employing Latent Blend for preserving the background is that, given a desired area mask, the less noisy foreground latent can be guided by the target text prompt $\Delta_\tau$. Meanwhile, the latent features outside the mask (the background) can be preserved. This blending ensures that, even if the latent feature within the edit area is modified, the background features stay consistent.

## J    EXPERIMENTAL DETAILS

For FateZero[3] (Qi et al., 2023), we employ prompt-to-prompt(Hertz et al., 2022) replace editing. To enhance the identity binding of the edited object, we set the self/cross replacement steps at 0.3 and

---

[3]https://github.com/ChenyangQiQi/FateZero

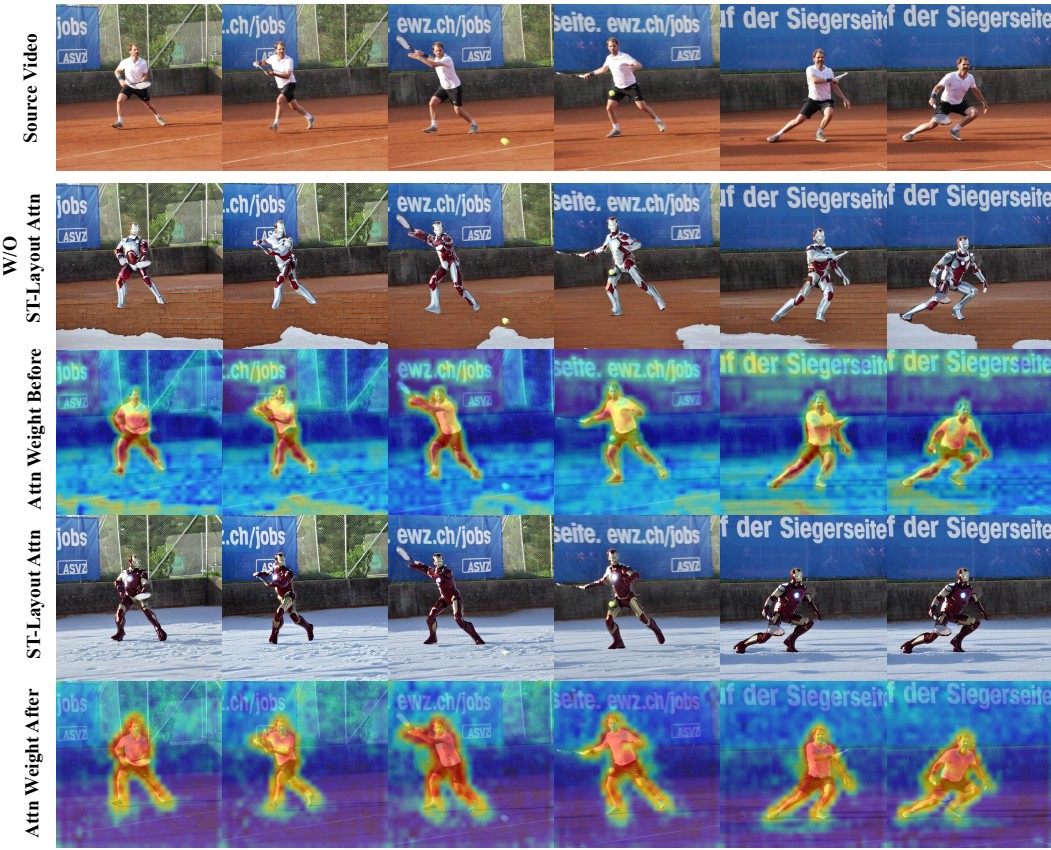

**Figure 17:** More frames ablation of ST-Layout Attn's effects on attention weight distribution.

the blending threshold at 0.7. In TokenFlow[4] (Geyer et al., 2023), we utilize SD editing and default to 4 keyframes for 16-frame videos. For other comparative methods like ControlVideo[5] (Zhang et al., 2023b) and Ground-A-Video[6] (Jeong & Ye, 2023) and DMT[7] (Yatim et al., 2024), we adhere to their default hyperparameter settings. To ensure fairness across all T2I-based methods compared, we re-implement ControlNet (Zhang et al., 2023a) on their codebases.

## K  LIMITATIONS.

First, although our method can achieve multi-grained editing of video, the generation quality is still limited by the base model since we are a training-free method. In scenarios where the generation prior to SD is not ideal, artifacts may occur in the editing results. Second, since our method is based on a T2I model, it struggles with large shape deformations and significant appearance changes. This limitation is inherent in zero-shot methods. A potential future direction is to incorporate motion priors from T2V generation models (Yang et al., 2024e) to handle such challenges.

---

[4] https://github.com/omerbt/TokenFlow
[5] https://github.com/YBYBZhang/ControlVideo
[6] https://github.com/Ground-A-Video/Ground-A-Video
[7] https://github.com/diffusion-motion-transfer/diffusion-motion-transfer

