# OpenReview forum: "VideoGrain: Modulating Space-Time Attention for Multi-Grained Video Editing"
_ICLR.cc/2025/Conference — ICLR 2025 Poster_

### Official Review · Reviewer_4zxu · 2024-10-29

**Soundness:** 2
**Presentation:** 3
**Contribution:** 3
**Rating:** 6
**Confidence:** 5

**Summary:**

This paper introduces ST-Modulator, a zero-shot framework for multi-grained video editing, enabling class-level, instance-level, and part-level modifications in diffusion models. To address semantic misalignment and feature coupling issues, ST-Modulator modulates space-time cross- and self-attention, allowing fine-grained text-to-region control and improved feature separation. Experimental results suggest that this approach performs competitively across benchmarks without additional parameter tuning, presenting a promising solution for multi-grained video editing.

**Strengths:**

- The paper clearly defines the novel task of multi-grained video editing, motivated by practical needs in video editing. By enabling edits at class-level, instance-level, and part-level, it broadens the range of possible video modifications to better meet real-world demands.
- The paper effectively uses visualizations to illustrate key points, such as in the *Analysis of why the diffusion model failed in instance-level video editing.* These visual aids provide intuitive insights into challenges with existing diffusion models and help clarify the motivation behind the proposed approach.
- The proposed Spatial-Temporal Layout-Guided Attention focuses on self-attention and cross-attention mechanisms to address key issues in text-to-region control and inter-region feature separation. This method effectively targets the core issues in multi-grained editing within diffusion models.
- The proposed method does not require parameter tuning, which makes it resource-efficient—a significant advantage for video editing applications where computational cost and tuning complexity are critical factors.

**Weaknesses:**

1. **Focus on Image Editing over Video-Specific Needs:** The approach seems more aligned with image editing, as it emphasizes spatial layout control. However, a key distinction in video editing is maintaining inter-frame consistency, which is crucial for coherent video results. While the Spatial-Temporal Layout-Guided Attention is introduced, the temporal aspect appears to receive less emphasis or detailed explanation.

2. **Lack of Clarification on Additional Control Signals:** The authors mention that the method is compatible with ControlNet conditioning, but it remains unclear if the cases shown in the paper require extra control inputs. There is no ablation study on control conditions to assess the necessity or influence of these signals. This raises questions about fairness in comparisons with other models like ControlVideo, where discrepancies in control conditions might lead to potentially biased results.

3. **Dependence on SAM-Track for Segmentation:** The method relies on SAM-Track for instance segmentation, yet no ablation study examines its impact. This dependency raises concerns: it is unclear if the observed performance gain primarily stems from using SAM-Track. For example, using SAM-Track segmentations as conditional input to other video editing models might yield similar improvements, suggesting a need for direct comparisons to validate the unique contributions of the proposed approach.

4. **Lack of Code or Algorithm Details:** The paper does not provide code or a detailed algorithm description. Open-sourcing the code could help address concerns such as those in Weaknesses 2 and 3 by allowing for transparent validation of the method’s assumptions, control conditions, and segmentation impacts.

**Questions:**

See weaknesses.

---

> ### Author Response · Authors · 2024-11-23
>
> >  **W1: Limited temporal focus**
>
> Thank you for your comment. Our ST-Layout Attn is designed as a full-frame approach to ensure inter-frame consistency. As shown in [[temporal focus](https://anonymous.4open.science/w/ST-Modulator-3EB3/#temporal%20focus)], per-frame ST-Layout Attn causes feature coupling on Iron Man, while the sparse-causal method results in flickering and misses Spider Man's blue details due to their limited receptive fields for positive/negative value selection across different layouts. In contrast, our full-frame ST-Layout Attn effectively preserves texture details and prevents flickering, achieving temporal consistency and layout integrity.
>
> >  **W2: ControlNet Clarification**
>
> 1. Our method utilizes ControlNet depth/pose conditioning in certain complex motion cases to ensure motion consistency. To address this, we have included an ablation study demonstrating the necessity and influence of ControlNet conditioning, as shown in [[ControlNet ablation](https://anonymous.4open.science/w/ST-Modulator-3EB3/#ControlNet%20ablation)].
>
> 2. For fairness, as noted in lines 427 and 818 of our manuscript, all T2I-based methods were equipped with the same ControlNet conditioning. Additionally, since ControlVideo and Ground-A-Video inherently use ControlNet, we reimplemented TokenFlow and FateZero within our framework with ControlNet conditioning.
>
> >  **W3: Dependence on SAM-Track for Segmentation**
>
> 1. Our method is not strictly dependent on SAM-Track masks. As shown in [[without sam-track results](https://anonymous.4open.science/w/ST-Modulator-3EB3/#Without%20SAM-Track)], we can cluster inversion self-attention features to get coarse layouts ([second column](https://anonymous.4open.science/w/ST-Modulator-3EB3/#Without%20SAM-Track)), and then enable robust multi-area editing ([third column](https://anonymous.4open.science/w/ST-Modulator-3EB3/#Without%20SAM-Track)). Even **without SAM-Track masks**, ST-Modulator produces **high-quality** multi-area editing results.
>
> 2. For certain scenarios involving multiple same-class instances (e.g., Figure 3(b), distinguishing the left man from the right man), **diffusion models inherently struggle to differentiate instances**. In these cases, instance-level information, such as SAM-Track masks, is crucial for accurate multi-grained editing.
>
> 3. Previous methods, such as Ground-A-Video and VideoP2P (equipped with SAM-Track instance masks), fail to perform effective multi-area editing.
>     - As seen in Figures 2 and 6, Ground-A-Video struggles with both instance-level and part-level editing tasks, even with grounding
>     information (e.g., text-to-bounding box), which is comparable to SAM-Track's masks.
>     - Also,  as seen in [[VideoP2P with SAM-Track masks](https://anonymous.4open.science/w/ST-Modulator-3EB3/#VideoP2P)], VideoP2P fails to achieve multi-area edits, even after inferring three times with SAM-Track masks.
>
> These comparisons indicate that while SAM-Track provides layout guidance, it does not guarantee successful edits. This further highlights the novelty and robustness of our approach.
>
> > **W4 : Code release**
>
> We have released our code at [anonymous link](https://anonymous.4open.science/r/ST-Modulator-code-D142) and will continue to update more interesting applications.

---

> ### Author Response · Authors · 2024-11-25
> **Look forward to your post-rebuttal feedback!**
>
> Dear Reviewer 4zxu,
>
> Thank you again for your insightful suggestions and comments. Since the discussion deadline is approaching, we are happy to provide any additional clarifications you may need.
>
> In our previous response, we carefully addressed your feedback by:
>
> 1. Adding a temporal focus ablation experiment in Appendix E and Figure 19.
>
> 2. Adding SAM-Track dependency experiments in Appendices A and B, using SAM-Track masks in VideoP2P and Ground-A-Video to evaluate whether other methods can achieve similar multi-grained results.
>
> 3. Adding a new example without SAM-Track to evaluate the gain of our ST-Layout Attention in Appendix B.
>
> 4. Adding a ControlNet ablation in Appendix F.
>
> 5. Releasing the code.
>
> We believe these new experiments and the updated manuscript effectively address your concerns. Please do not hesitate to reach out if there are any further clarifications or analyses we can provide.
>
> Thank you for your time and thoughtful feedback!
>
> Best regards,
>
> The Authors

---

> > ### Comment · Area_Chair_Mf75 · 2024-11-25
> > **Any further post-rebuttal feedback**
> >
> > Dear Reviewer 4zxu,
> >
> > Any further post-rebuttal feedback after checking the authors' replies?
> >
> > AC

---

> > > ### Author Response · Authors · 2024-11-26
> > > **Thank you!**
> > >
> > > Dear Reviewer 4zxu,
> > >
> > > We sincerely appreciate your positive feedback and are delighted that our responses have addressed your concerns.  Your support and inclination toward acceptance are truly encouraging, and we deeply value your thoughtful review and constructive comments.
> > > Thank you once again!

---

> > ### Comment · Reviewer_4zxu · 2024-11-27
> > **Thanks  for your response**
> >
> > Thank you for your detailed response, which resolved most of my confusion. I have already increased the score.

---

### Official Review · Reviewer_T7q7 · 2024-11-03

**Soundness:** 3
**Presentation:** 4
**Contribution:** 4
**Rating:** 8
**Confidence:** 5

**Summary:**

This paper presents a new task called multi-grained video editing,  which encompasses class, instance and fine-grained video editing. The empirical study shows that the key obstacles hindering multi-grained editing in diffusion models are text-to-region misalignment and feature coupling.  This work utilizes a pretrained T2I diffusion model to address these problems without any training. More Specifically ,the method modulates cross-attention for text-to-region control and self-attention for feature separation. Effectiveness has been proven by extensive experiments and convincing qualitative results.

**Strengths:**

1. This paper addresses a highly significant problem of editing granularity in video editing. It proposes a new and interesting task called multi-grained editing, which includes class, instance, and part-level editing. It allows flexible editing at any granularity and breaks the inherent feature coupling within diffusion models.
2. The empirical study is insightful, explaining why diffusion models are limited to class-level video editing.
3. The proposed method is novel and consistent with its motivation. It adjusts cross-attention for accurate attention weight distribution, enables text-to-region control, and modulates self-attention for feature separation. The modulation process is operated in a unified manner, and the overall framework is training-free.
4. The qualitative experiments are impressive, indicating that ST-Modulator surpasses previous approaches in multi-grained editing.

**Weaknesses:**

1. Since this work proposes a new and interesting task focused on multi-grained editing, a holistic evaluation would be beneficial, such as establishing a benchmark and developing new metrics to evaluate the performance of current methods on multi-grained video editing.
2. ST-Modulator focuses on editing videos at different spatial granularities, but the temporal length is limited to 16-32 frames. I wonder whether ST-Modulator can be extended to handle longer video sequences, such as those with hundreds of frames.
3. What is the difference between multi-grained editing and multi-attribute editing?

**Questions:**

Please kindly address the questions in the weakness section.

---

> ### Author Response · Authors · 2024-11-23
>
> >  **W1: Multi-grained Editing benchmark and metrics**
>
> We plan to establish a holistic multi-grained editing benchmark with 200 videos, covering class, instance, and part-level editing tasks. To evaluate instance- and part-level editing accuracy, we will mask out the ROI editing regions and employ GriT [1] for frame-wise object detection, measuring the proportion of frames where all required edits appear simultaneously.
>
>
> > **W2:  Video length**
>
> On an A100 GPU, our method can process up to ~200 frames. Additionally, we implemented sliced attention within ST-Layout Attn, significantly reducing memory requirements and extending the temporal range for longer videos.
>
> >  **W3:  Difference between Multi-Grained and Multi-Attribute Editing**
>
> Multi-grained editing extends beyond multi-attribute editing by addressing spatial-temporal granularity, with difficulty increasing as granularity becomes finer.
>
> Multi-attribute editing is confined to changes within the same category (e.g., man → Iron Man or changing clothing color). In contrast, multi-grained editing includes cross-category transformations, such as editing the right man into a monkey, which belongs to instance editing in multi-grained editing.
>
> [1] Wu et al., *GRIT: A Generative Region-to-Text Transformer for Object Understanding,* CVPR 2023.

---

> ### Author Response · Authors · 2024-11-25
> **Look forward to your post-rebuttal feedback!**
>
> Dear Reviewer T7q7,
>
> Thank you again for your insightful suggestions and comments. Since the discussion deadline is approaching, we are happy to provide any additional clarifications you may need.
>
> In our previous response, we carefully addressed your feedback regarding:
>
> 1. The metrics suitable for multi-grained video editing.
>
> 2. The maximum video editing length supported by ST-Modulator.
>
> 3. The difference between multi-grained and multi-attribute video editing.
>
> We sincerely appreciate your positive reviews. Please do not hesitate to reach out if there are any further clarifications or analyses we can provide.
>
> Thank you for your time and thoughtful feedback!
>
> Best regards,
>
> The Authors

---

> > ### Comment · Reviewer_T7q7 · 2024-11-27
> > **Nice work！**
> >
> > - The authors have addressed **all my concerns**. The visual performances are very impressive. The writing is easy to understand. It is the first work to achieve fine-controllable vide editing. I strong recommend it to Spotlight/Oral ! !
> >
> > - Video editing is a hot topic in video generation. How to handle various level objects is important research topic. This work presents lots of visual results and comparisons with previous work, claiming the superiority of proposed method. I hope the authors can add other reviewers' comments into paper and get the oral !!

---

### Official Review · Reviewer_XFkN · 2024-11-05

**Soundness:** 3
**Presentation:** 2
**Contribution:** 2
**Rating:** 6
**Confidence:** 4

**Summary:**

This paper presents a zero-shot approach that modulates attention mechanisms to achieve fine-grained control over video content. We enhance text-to-region control by amplifying each local prompt’s attention to its corresponding spatial disentangled region while minimizing interactions with irrelevant areas.  Extensive experiments demonstrate the method achieves state-of-the-art performance in real-world scenarios.

**Strengths:**

1. This is the first attempt at multi-grained video editing.
2. The results show the SOTA performance on existing benchmarks and real-world videos.
3. The method is intuitive and is easy to follow.

**Weaknesses:**

1. The readability and presentation of the manuscript need to be enhanced. For example, in Figure 4:
The "I" in "M_i^pos" denotes the i-th frame, whereas the "i" in "cross-attention modulation" appears to refer to the i-th token (e.g., "polar" is the 3rd token?). Readers should not have to guess what these symbols mean; it would be beneficial to clarify these points explicitly.
The "E" in "L × E" is undefined within the context of this paper, and Figure 2 does not utilize this information. Consequently, the shape information in Figure 4 could be omitted to avoid confusion.
It is unclear whether "p" in Figure 4 represents the original attention score. This should be clearly stated to avoid ambiguity.
2. The paper employs an additional model, SAM-Track, to perform instance segmentation, which might be unfair. If this is the case, one could directly use Grounding-DINO in conjunction with SAM-Track to accurately identify the edited area (mask) based on the provided text. Subsequently, methods like prompt-to-prompt or Video-P2P could be employed to control the editing process through attention maps, potentially achieving similar results.
3. A key concept of the paper is to enhance attention scores in relevant areas while suppressing them in irrelevant areas. However, it's worth noting that methods such as prompt-to-prompt already implement mechanisms to either reduce or increase attention scores for editing purposes. This aspect should be discussed more thoroughly to highlight the novelty and added value of the proposed approach.

**Questions:**

See weakness.

---

> ### Author Response · Authors · 2024-11-23
>
> Dear reviewer XFkN,
>
> Thank you for your valuable feedback. We have carefully considered your comments and suggestions to improve our paper. Below are our responses to your concerns.
>
> > **W1: Notations Error**
>
> We have made the following revisions to Figure 4:
> 1. We have distinguished the word index and frame index by using "w" and "i," respectively.
> 2. "p" indeed represents the original attention value before the modulating process in both self and cross attention.
> 3. We have removed the extraneous shape information and updated the definitions of all notations in the legend and caption of Figure 4.
>
> > **W2: Can VideoP2P perform multi-area editing with SAM-Track masks?**
>
> We further conducted experiments with VideoP2P, which indicates that it fails to edit multiple areas jointly, even when equipped with instance segmentation masks from SAM-Track and one-shot finetuning. Our method, ST-Modulator, is a zero-shot approach designed for multi-area editing in one denoising process.
>
> In our experiment, we first perform 16-frame one-shot tuning and then use attention replace and reweight in prompt-to-prompt editing to edit three areas jointly. In the editing stage, we replace the original cross-attention masks with SAM-Track instance masks and add Controlnet for fairness.
>
> We perform two experiments. The first is to edit three areas jointly, and we replace three subjects’ cross-attention masks with three areas’  SAM-Track masks,  then perform P2P edit in one denoising process. The second is to infer three times for separate editing of three regions, and we input three area masks sequentially.
>
> 1. **Jointly editing multiple areas at once.** As shown in [[videop2p with sam-track masks](https://anonymous.4open.science/w/ST-Modulator-3EB3/#VideoP2P)] first row, when inputting three instance masks jointly, VideoP2P was only able to modify the upper body of the left man into "Spiderman," with no effective edits on the right man or the trees. This is because the cross-attention weight distribution of VideoP2P is not accurate, as seen in [[videop2p cross attention weight visualization, top row](https://anonymous.4open.science/w/ST-Modulator-3EB3/#Difference)].
>
> 2. **Sequential editing based on separate masks.** Since VideoP2P failed to edit multiple areas in a single denoising process, we conducted a sequential test by providing separate masks to edit one area at a time. Specifically, we used the masks for the left man, right man, and trees as inputs sequentially. The results of the first and second edits were used as inputs for the second and third editing steps, respectively.
>
>     As shown in [[videop2p sequential editing results](https://anonymous.4open.science/w/ST-Modulator-3EB3/#Sequential)], sequential editing only succeeded in editing the left man’s upper body into "Spiderman" at first time. However, the second and third editing attempts, targeting the right man and the trees, failed. This is because the diffusion denoising process accumulates significant errors, and repeated inference exacerbates these errors, leading to blurred textures and diminished detail in the video.
>
> 3. Additionally, we measured the time and memory usage, as summarized in the table below. Compared to VideoP2P, our ST-Modulator is more efficient and supports multi-area editing simultaneously.
>
>     | Editing Stage       | Time (min) | GPU Mem (GB) | RAM (GB) |
>     |---------------------|------------|-----------------|----------|
>     | VideoP2P Tuning     | 16.50      | 18.52           | 5.41     |
>     | VideoP2P 1st Edit   | 4.14       | 24.55           | 5.61     |
>     | VideoP2P 2nd Edit   | 4.16       | 24.55           | 5.55     |
>     | VideoP2P 3rd Edit   | 4.15       | 24.55           | 5.67     |
>     | **ST-Modulator**    |  **3.83**    | **15.94**     |**4.42** |
>
> These joint and sequential editing experiments results demonstrate that VideoP2P cannot achieve multi-area editing, even when provided with SAM-Track instance masks.

---

> > ### Author Response · Authors · 2024-11-23
> >
> > > **W3: Difference between P2P and ST-Modulator**
> >
> > 1. Difference in increase/decrease attention scores:
> >     - P2P methods like Attention Reweight increase or decrease cross-attention scores by **multiplying** existing cross-attention weights with a factor (e.g., increasing by a factor of 4 or decreasing by 0.5). Note that this process occurs independently **after** the cross-attention computation, altering only the intensity, **without modifying the cross-attention weight distribution**.
> >
> >    - In contrast, our ST-Modulator increases and decreases attn scores **within** the cross-attention process. As depicted in formula 1 in manusript, we directly modify the scaled dot-product values of visual query features and text key embeddings when computing cross-attention. Specifically, we **add** positive scores (e.g., from the left man's visual features) and **subtract** negative scores (e.g., from irrelevant areas). Unlike P2P, this modulate process does not simply scale scores—we modulate the cross attention weight distribution, enabling text-to-multi regions control.
> >
> > 2. Difference in attention weight distribution. To further illustrate the differences, we visualized the cross-attention weights of VideoP2P and our ST-Modulator, as shown in [[cross-attn weight visualization](https://anonymous.4open.science/w/ST-Modulator-3EB3/#Difference)].
> >
> >     - VideoP2P (top row) fails to distribute attention weights effectively. For example, the weight for "Spiderman" is concentrated on the left man's upper body, resulting in an incomplete edit. While the weights for "polar bear" and "cherry blossoms" are weak and uneven, causing failed edits. This reflects the limitation of P2P methods, which only scale weights but cannot correct inaccurate attention distributions.
> >
> >     - In contrast, our ST-Modulator (bottom row) precisely aligns attention weights with the correct regions, enabling successful multi-area editing in one denoising process.

---

> ### Author Response · Authors · 2024-11-25
> **Look forward to your post-rebuttal feedback!**
>
> Dear Reviewer XFkN,
>
> Thank you again for your insightful suggestions and comments. Since the discussion deadline is approaching, we are happy to provide any additional clarifications you may need.
>
> In our previous response, we carefully addressed your feedback by:
>
> 1. Revising the notation error in Figure 4 and Section 3.4, clarifying the definition of "p," and correcting the word and frame indices.
>
> 2. Conducting experiments to evaluate whether VideoP2P can achieve similar results when equipped with SAM-Track masks, as shown in Appendix A.
>
> 3. Explaining the difference between P2P and our ST-Modulator, supported by visual evidence in Figure 9.
>
> We believe these new experiments and the updated manuscript effectively address your concerns. Please do not hesitate to reach out if there are any further clarifications or analyses we can provide.
>
> Thank you for your time and thoughtful feedback!
>
> Best regards,
>
> The Authors

---

> > ### Comment · Area_Chair_Mf75 · 2024-11-25
> > **Any further post-rebuttal feedback**
> >
> > Dear Reviewer XFkN,
> >
> > Any further post-rebuttal feedback after checking the authors' replies?
> >
> > AC

---

> > > ### Comment · Reviewer_XFkN · 2024-11-26
> > >
> > > Dear Authors,
> > >
> > > Thank you for your detailed response and the additional experiments. The supplementary results effectively addressed my concerns. Based on the response, I have increased my score accordingly.

---

> > > > ### Author Response · Authors · 2024-11-26
> > > > **Thank you!**
> > > >
> > > > Dear Reviewer XFkN,
> > > >
> > > > Thank you for your thoughtful review and for recognizing our efforts to address your concerns. We sincerely appreciate your questions, which allowed us to further clarify the novelty and contributions of our work. Thank you once again!

---

### Official Review · Reviewer_cwR6 · 2024-11-08

**Soundness:** 3
**Presentation:** 3
**Contribution:** 3
**Rating:** 6
**Confidence:** 3

**Summary:**

This paper introduces a new task called multi-grained video editing, which includes class-level, instance-level, and part-level editing, and proposes a zero-shot approach, ST-Modulator, to address the challenge of distinguishing distinct instances (e.g., "left man" and "right man") by modulating space-time attention mechanisms for precise, fine-grained control over video content—all without additional training.

**Strengths:**

- The paper is well-written and organized, making complex concepts easy to understand.
- The Spatial-Temporal Layout-Guided Attention method in this paper effectively addresses the challenge of precise, multi-grained video editing by modulating cross- and self-attention. This approach enables accurate text-to-region control and clear feature separation, allowing independent edits to specific subjects while avoiding unintended changes in other areas, especially in complex multi-subject scenes.
- Compared to other approaches, ST-Modulator achieves high computational efficiency, showing lower memory usage and faster processing times.

**Weaknesses:**

- Limited focus on background preservation.
- In Figures 5 and 6, most examples demonstrate edits that include modifications to the background along with the main subjects. Could the authors provide additional examples where the editing focuses solely on specific subjects, allowing the background to remain unchanged? This would help illustrate the method’s capability for selective edits in multi-subject scenes.

**Questions:**

- For part-level editing, is the method limited to adding objects, or can it also support modifications like changing the color of clothing or the color of an animal?

---

> ### Author Response · Authors · 2024-11-23
>
> Dear reviewer cwR6,
>
> Thank you for recognizing the key contributions and novelty of our work.
>
> > **W1: Solely edit on specific subjects, keep background preserved**
>
> We appreciate your insightful review highlighting the editing flexibility. Following your advice, we added examples showcasing our ability to edit specific subjects. As shown in [[solely editing examples, background unchanged](https://anonymous.4open.science/w/ST-Modulator-3EB3/#Soely%20Edit)], we have incorporated separate edits of the "left man," "right man,"  and joint editing of both subjects while keeping the background unchanged.
>
> > **W2: Modification examples in part-level editing**
>
> Thanks for your constructive comments. Our part-level editing is not limited to adding objects but also supports part-level modifications. We further validate our effectiveness in both human beings and animals.
>
> As shown in the human case in [[part-level modification examples](https://anonymous.4open.science/w/ST-Modulator-3EB3/#Part%20Level%20Modification)], our method can change the color of a gray shirt to blue, demonstrating part-level attribute editing. More complexly, our method can handle structural and texture modifications,  like modifying a half-sleeve shirt into a black suit.
>
>
> Also, as seen in the animal case in [[part-level modification examples](https://anonymous.4open.science/w/ST-Modulator-3EB3/#Part%20Level%20Modification)],  our method can change the head of a cat’s color from black to ginger, and alter the body’s color from black to ginger without altering the belt.

---

> > ### Comment · Reviewer_cwR6 · 2024-11-25
> > **Response**
> >
> > I appreciate the authors' response, as the additional results and clarifications have addressed my concerns and pleased to raise my rating. The qualitative results provided look promising, and suggest showcasing more of these background preservation cases in the revised paper to enhance its clarity and impact.

---

> > > ### Author Response · Authors · 2024-11-26
> > > **Thank you!**
> > >
> > > Dear reviewer cwR6,
> > >
> > > Thank you for your thoughtful review and for recognizing our efforts to address your concerns. We will further showcase more background preservation cases in the main figure to enhance clarity. We sincerely appreciate your consideration and insights. Thank you once again!

---

> ### Author Response · Authors · 2024-11-25
> **Look forward to your post-rebuttal feedback!**
>
> Dear Reviewer cwR6,
>
> Thank you again for your insightful suggestions and comments. Since the discussion deadline is approaching, we are happy to provide any additional clarifications you may need.
>
> In our previous response, we carefully addressed your feedback by:
>
> 1. Adding new examples focused solely on editing without background changes.
>
> 2. Adding more examples of humans and animals to demonstrate part-level modification abilities, such as color changes.
>
> We sincerely appreciate your suggestions, which have helped us expand the definition of part-level editing and better showcase selective editing capabilities.
>
> We believe these new experiments and the updated manuscript effectively address your concerns. Please do not hesitate to reach out if there are any further clarifications or analyses we can provide.
>
> Thank you for your time and thoughtful feedback!
>
> Best regards,
>
> The Authors

---

### Author Response · Authors · 2024-11-25
**General Response**

We thank all the reviewers for their valuable reviews. We also appreciate their recognition of the key contributions of our framework and the superiority of our results. Overall, we are encouraged that they find that:

1. **Contributions to the new task**: This paper proposes a new and interesting task called multi-grained editing, which includes class-level, instance-level, and part-level editing,  motivated by practical needs in video editing. (all reviewers)

2. The **novelty** of ST-Modulator: The proposed ST-Layout Attn is novel and effectively addresses the key challenges of  precise, multi-grained video editing by modulating cross- and self-attention.” (Reviewer cwR6, T7q7, 4zxu)

3. **Superior performance** of ST-Modulator: “The results show the SOTA performance and efficiency on existing benchmarks and real-world videos.” (Reviewer cwR6, XFkN, T7q7))

4. The merit of **training-free** framework: The proposed method does not require parameter tuning, a significant advantage for video editing (Reviewer cwR6, T7q7,4zxu)

5. The paper is well-written, and the empirical study is insightful, making complex concepts easy to understand.(Reviewer cwR6, T7q7, 4zxu)

According to our understanding, the main concern from reviewers is:

**Can other methods achieve similar results when equipped with SAM-Track instance masks?**

We conducted thorough experiments with VideoP2P equipped with SAM-Track masks (Figure 9) and Ground-A-Video (Figure 10, 11). The results show that both methods failed in multi-grained edits, even with instance masks. While SAM-Track provides layout guidance, it does not guarantee successful edits. Furthermore, as demonstrated in Appendix B, our method can achieve high-quality results even without SAM-Track.

Here, we have summarized the changes to our manuscript. Modified or added contents have been highlighted blue in the revised paper. Point-to-point responses were also included as a reply to each reviewer.

1. In **Appendix A** and **Figure 9**, we conducted experiments with VideoP2P equipped with SAM-Track masks and visualized cross-attn weight to illustrate the difference between P2P and ours. (Reviewer XFkN, 4zxu)

2. In **Appendix B** and **Figure 11**, we provide examples without sam-track masks and compare Ground-A-Video with instance conditions. (Reviewer 4zxu)

3. In **Appendix C**, we provided solely editing examples without changing the background(Reviewer cwR6)

4. In **Appendix D** and **Figure 1, 13,**, we expand part-level definition (Figure1 abd Line75) and add part-level modification examples. (Reviewer cwR6)

5. In **Figure 4** and **Section 3.4**, we revised the notations to clarify. (Reviewer XFkN)

6. In **Appendix E**, we provide the temporal focus of ST-Layout Attn (Reviewer 4zxu)

7. In **Appendix F**, we ablate the control condition signals. (Reviewer 4zxu)

8. We have already released the code at an anonymous link.

---

### Meta-Review · Area_Chair_Mf75 · 2024-12-19

**Metareview:**

After rebuttal and multiple rounds of discussion, all four reviewers unanimously agreed to accept this submission. One of the reviewers recommended it for a spotlight/oral presentation. From the anonymous website provided by the authors, we can see many promising examples. However, there are still some issues, e.g., temporal inconsistencies, jitter or overly smoothed backgrounds. So I recommend "Accept (poster)".

**Additional Comments On Reviewer Discussion:**

During the rebuttal process, three reviewers mentioned that the authors' responses resolved the issues they raised in the first round of review, and they raised their scores to 6. One reviewer maintained their original score of 8 and recommended the submission for a spotlight/oral presentation.

---

### Decision · Program_Chairs · 2025-01-22

Accept (Poster)